# Double Balloon Catheter (Plus Oxytocin) versus Dinoprostone Vaginal Insert for Term Rupture of Membranes: A Randomized Controlled Trial (RUBAPRO)

**DOI:** 10.3390/jcm11061525

**Published:** 2022-03-10

**Authors:** Eric Devillard, Fanny Petillon, Marion Rouzaire, Bruno Pereira, Marie Accoceberry, Céline Houlle, Lydie Dejou-Bouillet, Pamela Bouchet, Amélie Delabaere, Denis Gallot

**Affiliations:** 1Department of Obstetrics and Gynecology, CHU Clermont-Ferrand, 63000 Clermont-Ferrand, France; eric.devillard@gmail.com (E.D.); fpetillon@chu-clermontferrand.fr (F.P.); mrouzaire@chu-clermontferrand.fr (M.R.); maccoceberry@chu-clermontferrand.fr (M.A.); choulle@chu-clermontferrand.fr (C.H.); lbouillet@chu-clermontferrand.fr (L.D.-B.); pbouchet@chu-clermontferrand.fr (P.B.); adelabaere@chu-clermontferrand.fr (A.D.); 2Department of Obstetrics and Gynecology, Lucien Hussel Hospital, 38209 Vienne, France; 3Biostatistics Unit (DRCI), CHU Clermont-Ferrand, 63000 Clermont-Ferrand, France; bpereira@chu-clermontferrand.fr; 4“Translational Approach to Epithelial Injury and Repair” Team, Faculty of Medicine, Université Clermont-Auvergne, CNRS 6293, INSERM 1103, GReD, 63000 Clermont-Ferrand, France

**Keywords:** premature rupture of membranes, labor induction, unfavorable cervix, cervical ripening balloon, nulliparous

## Abstract

Background: The aim of this study is to demonstrate that a double balloon catheter combined with oxytocin decreases time between induction of labor and delivery (TID) as compared to a vaginal dinoprostone insert in cases of premature rupture of membranes at term. Methods: This is a prospective, randomized, controlled trial including patient undergoing labor induction for PROM at term with an unfavorable cervix in Clermont-Ferrand university hospital. We compared the double balloon catheter over a period of 12 h with adjunction of oxytocin 6 h after catheter insertion versus dinoprostone vaginal insert. After device ablation, cervical ripening continued only with oxytocin. The main outcome was TID. Secondary outcomes concerned delivery mode, as well as maternal and fetal outcome, and were adjusted for parity. Results: 40 patients per group were randomized. Each group had similar baseline characteristics. The study failed to demonstrate reduced TID (16.2 versus 20.2 h, ES = 0.16 (−0.27 to 0.60), *p* = 0.12) in the catheter group versus dinoprostone except in nulliparous women (17.0 versus 26.5 h, ES = 0.62 (0.10 to 1.14), *p* = 0.006). The rate of vaginal delivery <24 h significantly increased with combined induction (88.5% versus 66.6%, *p* = 0.03). No statistical difference was observed concerning caesarean rate (12.5% versus 17.5%, *p* > 0.05), chorioamnionitis (0% versus 2.5%, *p* = 1), postpartum endometritis, or maternal or neonatal outcomes. Procedure-related pain and tolerance to devices were found to be similar for the two methods. Interpretation: The double balloon catheter combined with oxytocin is an alternative for cervical ripening in case of PROM at term, and may reduce TID in nulliparous women.

## 1. Introduction

The premature rupture of membranes (PROM) at term, a complication in 8% of pregnancies [1], is associated with risk of chorioamnionitis and neonatal sepsis, which increase with PROM duration [2,3]. Spontaneous labor occurs in 60–70% of these patients within 24 h [2,3,4]; however, when no effective uterine contraction occurs, induction of labor (IOL) is the optimal strategy for women with PROM at term, according to recommendations by French and American Colleges of Obstetricians and Gynecologists [5,6].

Prostaglandins and oxytocin are frequently used for cervical ripening in cases of PROM(7), and are reported to be of similar efficacy when there is an unfavorable cervix [7]. Mechanical induction using a balloon is generally considered as effective as vaginal dinoprostone, although less effective than low-dose vaginal misoprostol, despite improved levels of safety. Studies comparing a mechanical device (Foley catheter) for IOL in cases of PROM versus prostaglandins or oxytocin have reported similar time intervals from induction to delivery, and no differences concerning maternal or neonatal infections have been reported, with the exception of one retrospective study that revealed quicker deliveries associated with a Foley catheter [8,9,10,11]. In multiparous women with intact membranes, the simultaneous use of a cervical ripening balloon and oxytocin led to higher rates of delivery within 24 h, and a shorter induction-to-delivery interval without adverse maternal or neonatal outcomes [12]. Previously, two trials conducted in women with PROM at 34 or more gestational weeks reported that the combined use of a Foley catheter and oxytocin was not found to shorten the time to delivery, compared with oxytocin alone [9,10]. In one of these trials, an increased risk of intra-amniotic infection was found.

Since the optimal method for IOL in cases of PROM at term is currently unknown, the main objective of this study was to determine whether the use of a double balloon catheter combined with oxytocin would lead to a reduction in time between IOL and delivery (TID) when compared to a vaginal dinoprostone insert. The secondary objective was to compare maternofetal outcomes for each induction strategy, and to investigate the association between parity and choice of strategy.

## 2. Materials and Methods

This study was a prospective, monocentric, randomized, controlled clinical trial with two parallel arms comparing induction via double balloon catheter (plus oxytocin) with vaginal dinoprostone insert, in cases of PROM at term.

### 2.1. Participants

Women with a live, singleton gestation at term (37 or more weeks of gestation) with PROM (clinical amniotic fluid leakage and/or positive IGFBP1 test), an unfavorable cervical examination (Bishop score < 6), and no contraindication to labor who presented at the University Hospital of Clermont-Ferrand, France between February 2018 and March 2019, were approached for study participation. The hospital had around 3800 deliveries per year, and the rate of induction was 24%. Patients underwent spontaneously rupture of membranes at least 12 h before randomization.

Maternal exclusion criteria included those in active labor, suspected intraamniotic infection, detection of group B Streptococcus on any vaginal or urinary sample during the current or any previous pregnancy, placental abruption or significant hemorrhage, any prior uterine surgery including caesarean delivery, any contraindication to vaginal delivery, human immunodeficiency virus, or herpetic genital lesions. Fetal exclusion criteria were non-cephalic presentation, severe fetal anomalies, intrauterine fetal demise, growth restriction < 3rd percentile with Doppler anomalies, and non-reassuring fetal heart rate (FHR) tracing.

### 2.2. Study Procedures

Eligible women who gave written informed consent were enrolled and randomized. Randomization was conducted using Research Electronic Data Capture (REDCap) software, and was carried out in random-sized blocks with stratification on parity (nulliparous versus parous) [13].

Participants were randomly allocated to the double balloon catheter (plus oxytocin) or vaginal dinoprostone insert groups. A course of prophylactic antibiotics—amoxicillin, or clindamycin in case of allergy—was administered upon recruitment and up to the time of delivery to prevent chorioamnionitis, as recommended by French national guidelines [1]. The double balloon device used was the Cook^®^ Cervical Ripening Balloon, (Cook Medical Europe, Co. Limerick, Ireland, reference: J-CRBS-184000). The catheter was inserted following the manufacturer’s instructions [14]. Each balloon was filled with 80 mL of saline solution, and the device remained in place for 12 h. Oxytocin was started six hours after device insertion, with epidural analgesia following patient wishes. Oxytocin was continued according to uterine contractions after catheter loss or removal. In the second group, the vaginal dinoprostone insert (Propess^®^, Ferring SAS, Gentilly, France) was inserted for a maximum of 24 h. In cases where the insert was lost by itself in the first 12 h and the patient had no contractions or continued to present an unfavorable cervix, another vaginal system was placed for a maximum further 24 h. Oxytocin could then be administered 30 min after removal of the vaginal system, with or without epidural analgesia, as per patient request. Patient pain levels were assessed every six hours until device removal by a midwife, who also monitored temperature and blood pressure in accordance with the recommendations [15]. The management of obstructed labor, FHR abnormalities and final delivery method was at the discretion of the physician on duty. After delivery, placenta samples were collected, and maternal satisfaction was recorded before discharge. Data regarding potential maternofetal infections were recorded until the end of hospitalization, and used in the analysis. No additional visits after discharge were scheduled. The nature of the intervention rendered the blinding of physicians, midwives, or patients impossible. The protocol was published on the BMJ open website [16], and the main study stages are described in Figure 1.

Outcome data were either documented on an ongoing basis by the labor and delivery team, or obtained from medical records by research personnel not involved with data analysis. The primary outcome was the period of time between induction (time of induction device insertion) and delivery (time of birth). Secondary outcomes included time between PROM and the start of induction, delivery rate within the first 24 h, duration of induction device placement, spontaneous or assisted vaginal delivery rate, rate of caesarean section and indications, postpartum hemorrhage rate (blood loss > 500 mL), Bishop score or measure of dilatation on catheter loss or removal, rate of balloon expulsion within 12 h of placement, rate of oxytocin cessation, balloon or vaginal dinoprostone insert removal due to suboptimal FHR, epidural analgesia rate, time taken to achieve active labor and full dilatation, pain levels assessed using the Visual Analog Scale of Pain Intensity (VASPI) at time of placement, every 6 h and after removal, uterine hyperstimulation rate, fever during labor rate, rate of clinical chorioamnionitis(defined as fever and combination of fetal tachycardia, or uterine contraction, or purulent fluid from the cervical os, or abdominal pain), rate of materno-fetal infection, endometritis, histological chorioamnionitis, and positive bacteriological culture. Neonatal outcomes included birth weight, lactates, rate of Apgar score <7 at 5 min, umbilical artery pH < 7.15, and neonatal intensive care unit admission.

### 2.3. Statistical Analysis and Sample Size Calculations

Sample size estimation was based on data from our center and from results reported by Mackeen et al. [10]. To highlight a clinical and relevant absolute difference of 9 h, 26 patients per group were needed for a two-sided type I error of 5% and a statistical power of 90%. However, to ensure satisfactory statistical power for secondary outcomes, 40 patients per group were required.

Data storage and management were performed following international guidelines. Results from intermediate analysis and all records concerning transfer of participating patients or their newborns to intensive care or reanimation units were examined by an independent Data Monitoring Safety Committee. Intermediate safety analyses were conducted for caesarean section and chorioamnionitis rates.

All analyses were conducted with Stata software (version 13, StataCorp, College Station, TX, USA), in accordance with the International Conference on Harmonization Good Clinical Practice guidelines. All statistical tests were performed with a type I error at 5%, and primary analysis was based on intention to treat (ITT). Continuous variables were presented as means and SD, or as medians and quartiles [interquartile range], according to the statistical distribution. Normality was studied using the Shapiro–Wilk test. A comparison of the primary outcome between randomized groups was performed using the non-parametric Mann–Whitney test, as t-test assumptions were not met. Homoscedasticity was checked using the Fisher–Snedecor test. The result was also expressed using effect-size (ES) and 95% confidence interval (95% CI) after logarithmic transformation. The primary outcome was also treated as censored data associated with a favorable outcome (uncomplicated delivery, without caesarean section). The estimation was carried out using the Kaplan–Meier method, and comparison between randomized groups using the log-rank test.

Student’s *t*-test and Mann–Whitney test were applied for other quantitative parameters, with Chi-squared and Fisher’s exact tests used for categorical parameters. When appropriate, results were expressed using absolute differences and 95% confidence intervals.

On the basis of clinical relevance and European Medicines Agency and Consolidated Standards of Reporting Trials recommendations, subgroup analysis according to parity was performed after investigation of the interaction parity × randomization group.

## 3. Results

From February 2018 to March 2019, we randomized 80 patients, with 40 allocated to each group. Over this period, the refusal rate was 27%, with reason for refusal related to randomization. A flow chart presenting patient recruitment is shown in Figure 2.

There were no withdrawals or patients lost to follow up, and both groups were similar with regard to demographic and antenatal characteristics (Table 1).

All patients received the device allocated by randomization, with the exception of one patient, who received a double balloon and spontaneously went into labor immediately after randomization. The median interval between PROM and IOL was similar between the two groups (25.9 [22.8; 29.4] versus 26.8 [24.0; 29.4] h, *p* = 0.46) (Table 2).

Using intention to treat, the study failed to demonstrate reduced TID (16.2 versus 20.2 h, ES = 0.16 (−0.27 to 0.60), *p* = 0.12) but treatment by double balloon catheter (plus oxytocin) was found to be associated with a significantly higher rate of delivery <24 h (90% versus 57.5%, absolute difference = 33% (15 to 50), *p* = 0.001) and vaginal delivery <24 h (88.5% versus 66.6%, absolute difference = 22% (3 to 41), *p* = 0.03) (Figure 3).

In nulliparous women ripened using a double balloon catheter (plus oxytocin), the study found reduced TID (17.0 versus 26.5 h, ES = 0.62 (0.10 to 1.14), *p* = 0.006), a significantly higher rate of delivery <24 h (89.6% versus 41.4%, absolute difference = 48% (27 to 69), *p* = 0.001), vaginal delivery <24 h (87.5% versus 50%, absolute difference = 38% (13 to 62), *p* = 0.01), and shorter induction to active labor time (10.0 [7.5; 12.5] versus 14.7 [9.7; 23.6] h, ES = 0.69 (0.03 to 1.31), *p* = 0.03). However, the reduction in delay between induction and full cervical dilatation did not reach significance (14.5 [12.2; 19.1] versus 19.9 [13.1; 29.5] h, ES = 0.45 (−0.13 to 1.02), *p* = 0.06) (Table 2). Ripening device removal or oxytocin discontinuation for abnormal FHR rates were similar between the two groups, as were rates of uterine hyperstimulation and caesarean section (12.5% versus 17.5%, *p* = 0.75). Despite oxytocin quantities being significantly higher in the double balloon catheter (plus oxytocin) group, no differences were found in postpartum hemorrhage rates (Table 3) or neonatal outcomes (Table 4).

We observed no postpartum endometritis, only one materno-fetal infection, and no significant differences concerning clinical, bacteriological, or histological chorioamnionitis (Table 5).

Device placement was significantly less painful for the dinoprostone vaginal insert (VASPI: 4.6 ± 2.9 versus 2.9 ± 2.5, ES = 0.58 (0.10 to 1.06), *p* = 0.02). However, after the placement phase, reported pain levels were significantly lower in the double balloon catheter (plus oxytocin) group. A majority of patients responded positively to the question “Would you agree to use the same cervical ripening device during a future delivery?” (Table 6).

## 4. Discussion

The RUBAPRO trial failed to demonstrate that the association of a double balloon catheter with oxytocin decreased TID, compared to a vaginal dinoprostone insert except in nulliparous women, for whom a difference of 9 h was observed. Delivery <24 h and vaginal delivery <24 h rates, however, were increased in the double balloon catheter (plus oxytocin) group for the entire study population. We observed no differences in caesarean delivery, or maternal or neonatal infection rates, following the systematic administration of antibiotic prophylaxis.

A double balloon catheter and oxytocin combined appeared more efficient for nulliparous women, who were in the majority in both groups (29/40 patients in each group), in line with other trials studying ripening of PROM at term [8,10,17]. The few studies that have investigated the efficiency and safety of IOL for PROM using mechanical devices, at or near term, describe similar TID when compared to other methods [7,8,9,10], with the exception of Mackeen et al. [17], who reported significantly decreased TID when comparing the Foley catheter versus misoprostol in a retrospective bicentric study.

In other indications, however, several authors have compared IOL by double balloon catheter with adjunction of oxytocin versus dinoprostone or oxytocin [18,19]. They observed lower TID, and a higher rate of delivery <24 h in the group with combined catheter and oxytocin. A recent meta-analysis has also demonstrated that simultaneous use of oxytocin with a Foley catheter could shorten induction to delivery time and increase deliveries within 12 to 24 h [20]. The adjunction of oxytocin is likely to favor a synergistic action. As concomitant administration of prostaglandin and oxytocin is strictly forbidden to prevent the risk of tachysystolia or uterine hyperstimulation [21], oxytocin may be administrated earlier when used in combination with mechanical devices, positively impacting the period of time before birth. Finally, the combination of catheter plus oxytocin has not been shown to impact rates of caesarean section, postpartum hemorrhage, or neonatal complications [19,20,22].

Observation of systematic antibiotic prophylaxis and exclusion of group B streptococcus patients revealed no differences concerning maternal or fetal infection. Studies generally report no impact of mechanical devices on infectious complications even in cases of pre-labor membrane rupture [11,17,23,24]. Mackeen et al. highlighted an increased risk of clinical chorioamnionitis with the use of a Foley catheter and oxytocin versus oxytocin alone (8% versus 0%, *p* < 0.01) [10]. In this study, 30% of patients presented vaginal portage of group B streptococcus, which is an independent infectious risk factor in cases of PROM. Moreover, antibiotics were only administered in group B identified patients or to those with clinically suspected intraamniotic infection, and as expected, histological chorioamnionitis was more frequent than clinical chorioamnionitis. This is a current finding in a PROM context, as membrane inflammation greatly contributes both to rupture and entry into labor [25,26].

Patients were interviewed about their childbirth experience focusing on pain management and satisfaction [27]. Boyon et al. also observed a higher frequency of VASPI >4 during prostaglandin use when compared with a double balloon catheter [28]. Concordant results have been described by Lim et al., who found that women were equally satisfied with both methods [29].

No previous study has reported on the use of a double balloon catheter or compared use of a mechanical device with vaginal dinoprostone insert in a PROM-related context. Additional studies are required to support our results, notably concerning the role of parity, but also to investigate women’s satisfaction concerning their experience of IOL. Mechanical device safety should also be confirmed prior to extend use to group B streptococcus positive patients presenting with PROM.

The main limitations of this study concern the sample size and monocentric design. In addition, the nature of the intervention made it impossible to ‘blind’ physicians, midwives, or patients. To compensate for this absence of blinding, we chose to use an objective primary outcome. We initially estimated the sample size using a clinical and relevant absolute difference of 9 h, although our trial finally revealed an overall difference of 4 h except for nulliparous women (more than 9 h). A larger study would be required to address questions concerning maternal or neonatal infections.

## 5. Conclusions

Our study demonstrated that a double balloon catheter combined with oxytocin could be an alternative to a dinoprostone vaginal insert for cervical ripening in cases of PROM with unfavorable cervix at term. This combination was associated with significantly higher delivery <24 h or vaginal delivery <24 h rates, and may reduce TID in nulliparous women.

## Figures and Tables

**Figure 1 jcm-11-01525-f001:**
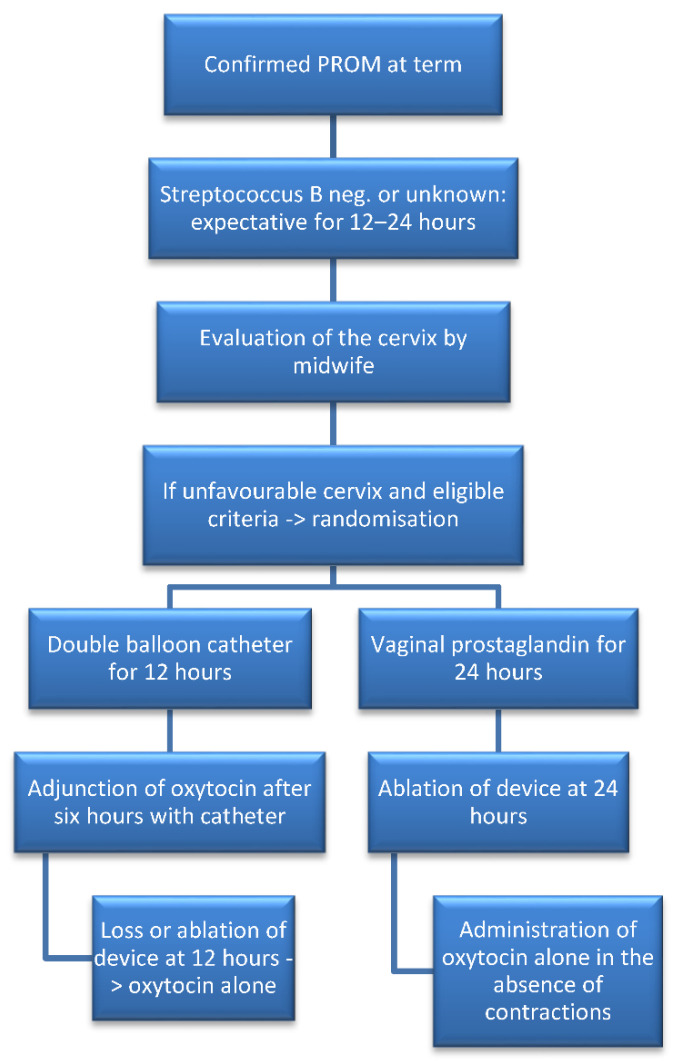
Flowchart describing the main stages of the study.

**Figure 2 jcm-11-01525-f002:**
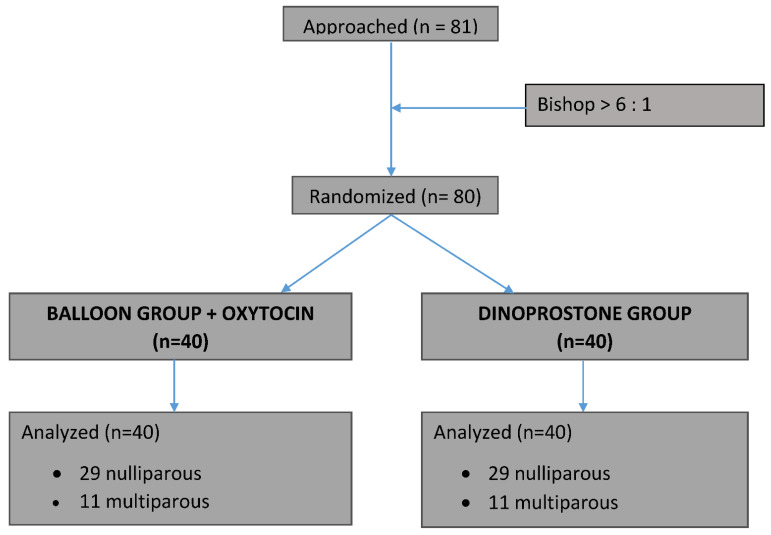
Flowchart of randomization into treatment groups.

**Figure 3 jcm-11-01525-f003:**
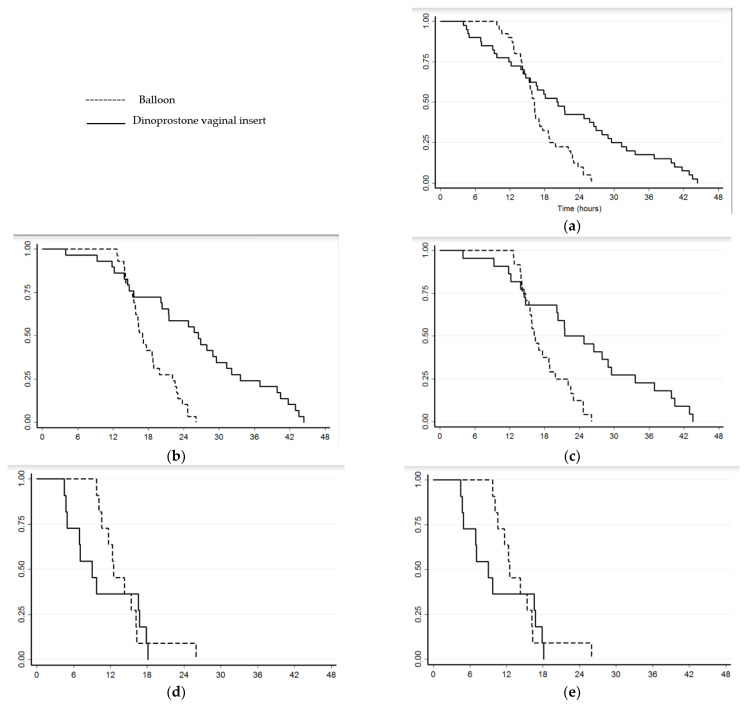
Kaplan–Meier survival curves illustrating induction to delivery time: (**a**) for entire study population; (**b**) for nulliparous and all deliveries modes; (**c**) for nulliparous and vaginal delivery; (**d**) for multiparous and all deliveries modes; (**e**) for multiparous and vaginal delivery.

**Table 1 jcm-11-01525-t001:** Demographic and antenatal characteristics.

Variable	Double Balloon Plus Oxytocine (*n* = 40)	Vaginal Dinoprostone Insert (*n* = 40)
Maternal age in y Median [IQR]	27.5 [24.9–30.6]	27.7 [25.4–30.4]
Parity		
Nulliparous	29 (72.5%)	29 (72.5%)
Parous	11 (27.5%)	11 (27.5%)
Maternal prepregnancy BMI in kg/m² Median [IQR]	23.3 [20.6–28.4]	24.6 [22–27.3]
Associated pregnancy pathologies	7 (17.5%)	9 (22.5%)
Diabetes mellitus	5 (12.5%)	6 (15%)
HTN	0	0
Preeclampsia	0	0
IUGR	0	0
Other	2 (5%)	4 (10%)
Gestational age at PROM in w, Median [IQR]	39.4 [38.2–40.4]	39.3 [38.5–40.3]
Bishop score at randomization Mean ± SD	3.5 ± 1.1	3.5 ± 1.3

BMI: body mass index; HTN: hypertension; IUGR: intrauterine growth restriction; IQR: interquartile range; SD: standard deviation.

**Table 2 jcm-11-01525-t002:** Labor and delivery characteristics.

Variable	Double Balloon Plus Oxytocine (*n* = 40)	Vaginal Dinoprostone (*n* = 40)	*p* Value
Time PROM to IOL in h Median [IQR]	25.9 [22.8–29.4]	26.8 [24–29.4]	0.46
Time IOL—loss ou removal of device in h, Median [IQR]	8.6 [5.0–12.0]	13.0 [6.6–18.4]	0.02
Time IOL—active labor phase in h, Median [IQR]	10 [6.8–12.5]	13.4 [8–19.8]	0.03
Nulliparous	10 [7.5–12.5]	14.7 [9.7–23.6]	0.03
Parous	9.4 [5.5–12.5]	9.9 [4.3–15.3]	0.60
Time IOL—full dilatation in h Median [IQR]	13.9 [11.8–18.3]	16.6 [9.3–24]	0.45
Nulliparous	14.5 [11.9–19]	19.9 [13–29.5]	0.06
Parous	12.3 [10.2–15.8]	8.7 [4.9–16.6]	0.22
Time IOL—delivery, h Median [IQR]	16.2 [14–19.4]	20.2 [12–30.4]	0.12
Nulliparous	17 [15.3–22]	26.5 [15.5–33.6]	0.006
Parous	12.6 [10.6–16.2]	9.0 [4.9–16.8]	0.19
Time IOL—vaginal delivery, h Median [IQR]	15.8 [13.8–18.9]	17.8 [9.7–27.9]	0.48
Nulliparous	16.3 [14.4–21]	23.1 [14.4–33.6]	0.06
Parous	12.6 [10.6–16.2]	9.0 [4.9–16.8]	0.19
Rate of IOL—delivery <24 h	36 (90%)	23 (57.5%)	0.001
Nulliparous	26 (89.6%)	12 (41.4%)	0.001
Parous	10 (90.9%)	11 (100%)	1.00
Rate of IOL—vaginal delivery <24 h	31 (88.5%)	22 (66.6%)	0.03
Nulliparous	21 (87.5%)	11 (50%)	0.01
Parous	10 (90.9%)	11 (100%)	1.00

IOL: Induction of labor; IQR: interquartile range.

**Table 3 jcm-11-01525-t003:** Maternal outcomes.

Variable	Double Balloon Plus Oxytocine (*n* = 40)	Vaginal Dinoprostone (*n* = 40)	*p* Value
Oxytocin use during labor in UI, Mean (± SD)	2.74 ± 3.22	1.33 ± 1.73	0.002
Abnormal FHR rate leading to:			
Ripening device removal	0	0	1.00
Oxytocin discontinuation	6 (15%)	3 (7.5%)	0.48
Uterine hyperstimulation	2 (5%)	1 (2.5%)	1.00
Fever during labor	1 (2.5%)	1 (2.5%)	1.00
Epidural use	40 (100%)	38 (95%)	0.49
Delivery mode			0.78
Vaginal delivery	35	33	
Spontaneous	30 (75%)	27 (67.5%)	
Extraction	5 (12.5%)	6 (15%)	
Caesarean section	5 (12.5%)	7 (17.5%)	
Caesarean indications	5	7	
Failure of induction	1 (20%)	1 (14.5%)	
Failure of dilatation progress	4 (80%)	5 (71%)	
Nonreassuring FHR	0	1 (14.5%)	
Postpartum hemorrhage *	5 (12.5%)	6 (15%)	0.75

SD: Standard Deviation. * defined as blood loss ≥500 mL after delivery.

**Table 4 jcm-11-01525-t004:** Neonatal outcomes.

Variable	Double Balloon Plus Oxytocine (*n* = 40)	Vaginal Dinoprostone (*n* = 40)	*p* Value
Birth weight, g Median [IQR]	3152.5 [2922–3542]	3275 [3047–3505]	0.36
5-min Apgar score <7	2 (5%)	1 (2.5%)	1.00
Umbilical artery pH Mean (±SD)	7.23 ± 0.07	7.24 ± 0.08	0.44
Umbilical artery BE Mean (±SD)	−4.10 ± 2.11	−4.89 ± 2.95	0.19
Lactates, mmol/L Mean (±SD)	4.15 ± 1.84	4.22 ± 1.59	0.87
NCIU Admission	1 (2.5%)	2 (5%)	1.00

BE: Base Excess; NCIU: Neonatal Intensive Care Unit; IQR: interquartile range.

**Table 5 jcm-11-01525-t005:** Materno-fetal infectious outcomes.

Variable	Double Balloon Plus Oxytocine (*n* = 40)	Vaginal Dinoprostone (*n* = 40)	*p* Value
Clinical chorio-amnionitis	0	1 (2.5%)	1.00
Bacteriological chorioamnionitis	0	3 (7.5%)	0.24
Histological	6 (15%)	5 (12.5%)	0.56
chorio-amnionitis			
funisitis	3	0	
Postpartum endometritis	0	0	
Materno-fetal infection			
No	38 (95%)	39 (97.5%)	1.00
Probably	1 (2.5%)	1 (2.5%)	
Confirmed	1 (2.5%)	0	

**Table 6 jcm-11-01525-t006:** Ripening device tolerance.

Variable	Double Balloon Plus Oxytocine (*n* = 40)	Vaginal Dinoprostone (*n* = 40)	*p* Value
VASPI at insertion device, Mean (±SD)	4.6 ± 2.8	2.9 ± 2.5	0.02
Nulliparous	4.7 ± 2.9	3.3 ± 2.6	0.07
Parous	3.6 ± 2.5	1.4 ± 1.5	0.13
VASPI at H6, Mean (±SD)	3.8 ± 2.9	5.8 ±2.9	0.04
Nulliparous	3.7 ± 2.9	5.6 ± 3.0	0.08
Parous	4.4 ± 2.7	7.0 ± 0.0	0.10
VASPI at H12, Mean (±SD)	2.5 ± 3.1	5.7 ± 2.9	0.008
VASPI at H18, Mean (±SD)		6.2 ± 1.3	
VASPI at H24, Mean (±SD)		5.1 ± 4.1	
Acceptability			
Yes	37 (92.5%)	36 (87.5%)	0.84
No	3 (7.5%)	3 (7.5%)	
No data	0	1 (2.5%) (il manque 1)	

VASPI: Visual Analog Scale of Pain Intensity; SD: Standard Deviation.

## Data Availability

Not applicable.

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
