# Peer review of "Double Balloon Catheter (Plus Oxytocin) versus Dinoprostone Vaginal Insert for Term Rupture of Membranes: A Randomized Controlled Trial (RUBAPRO)"

_jcm, 2022, doi:10.3390/jcm11061525_

Round 1

Reviewer 1 Report

The paper concludes that a double balloon catheter combined with oxytocin could be an alternative to a dinoprostone vaginal insert (Propess) for cervical ripening in cases of Term PROM. This combination was associated with significantly higher delivery <24h or vaginal delivery<24h rates and may reduce Time induction-delivery only in nulliparous women, without increasing feto-maternal infections (in a pupulation of women treated with antibiotic prophylaxis). The scientific accuracy is adequate.
The weaknesses of this study are the small sample size and the methods section that should be better explained. The strengths of this study is the RCT design and the accuracy of the conduction of the study. 

Please review the points below:

Introduction: 

line 54 page 2: Please explain that the FOLCROM TRIAL "Mackeen AD et al. Obstet Gynecol. 2018 Jan; who randomizet about 200 women at Foley + Oxitocin and Oxitocin alone and found that the use of a transcervical Foley catheter in addition to oxytocin does not shorten the time to delivery compared with oxytocin alone, but increase the incidence of intraamniotic infection (8% compared with 0%, P<.01). Please include this important finding in the manuscript. 

Material and methods: 

page 2, line 68: please explain the number of deliveries/year and the rate of induction of the Hospital.

page 2, line 85: Why antibiotic therapy was administred? Patients with GBS positivity were excluded from the randomization, thus it'a unclear the reason why antibiotic therapy was administred? This procedure IS NOT RECOMMENDED. Abuse of antibiotics may alter the microbiome of the newborn and select resistents germs.

page 2, line 98: The EMA organizations recommend the use of Oxitocyn after almost 6 hours after the Propess removal. Why the authors decided the Ox administration only 30 minutes after the removal. 

Page 4, line 124: Why authors used theese criteria? The Triple I criteria were the most used now and worldwide.

page 4, line 125: Please better explain "materno-fetal infection"? Please include in the placenta hystopatology analisys the presence of"funisitis" that is more sensitive for intramniotic infection than chorioamnionitis alone because of its aspecificity.

Results: succinted and weel described:

Discussion: 

page 10, line 241:  Please clarify the indication of use of antibiotics in patients screened and negative for GBS? Is the prophylactic use of antibiotics performed in order to minimize the risk of infection due to the Baloon in women with PROM? This concept must been speculated in the discussion section.

page 11, line 263: About this concept please cite the consequences of the systematic use of antibiotics in women without indication. (Microbioma, Multi resistance...)

Author Response

Response to Reviewer 1 Comments

The paper concludes that a double balloon catheter combined with oxytocin could be an alternative to a dinoprostone vaginal insert (Propess) for cervical ripening in cases of Term PROM. This combination was associated with significantly higher delivery <24h or vaginal delivery<24h rates and may reduce Time induction-delivery only in nulliparous women, without increasing feto-maternal infections (in a pupulation of women treated with antibiotic prophylaxis). The scientific accuracy is adequate.

The weaknesses of this study are the small sample size and the methods section that should be better explained. The strengths of this study is the RCT design and the accuracy of the conduction of the study.

  • This is a pilot study prior to a larger scale study -RUBAPROII- with a surrogate endpoint.

1- Please explain that the FOLCROM TRIAL "Mackeen AD et al. Obstet Gynecol. 2018 Jan; who randomizet about 200 women at Foley + Oxitocin and Oxitocin alone and found that the use of a transcervical Foley catheter in addition to oxytocin does not shorten the time to delivery compared with oxytocin alone, but increase the incidence of intraamniotic infection (8% compared with 0%, P<.01). Please include this important finding in the manuscript. 

  • A sentence was added in the introduction lines 54-55. This point is further discussed lines 303-313.

2-Please explain the number of deliveries/year and the rate of induction of the Hospital.

  • The number of deliveries/year and the rate of induction of the Hospital were added in the revised manuscript (lines 73-74).

3- Why antibiotic therapy was administred? Patients with GBS positivity were excluded from the randomization, thus it'a unclear the reason why antibiotic therapy was administred? This procedure IS NOT RECOMMENDED. Abuse of antibiotics may alter the microbiome of the newborn and select resistents germs.

  • According to french national guidelines « In the absence of spontaneous labor within 12hours of rupture, antibiotic prophylaxis could reduce the risk of maternal intrauterine infection but not of neonatal infection (LE3). Its use after 12hours of rupture in term prelabor rupture of the membranes is therefore recommended (Grade C). When antibiotic prophylaxis is indicated, intravenous beta-lactams are recommended (Grade C). »

[1] Term Prelabor Rupture of Membranes: CNGOF Guidelines for Clinical Practice. M-V Senat, T Schmitz, H Bouchghoul, C Diguisto, A Girault, S Paysant, J Sibiude, L Lassel, L Sentilhes PMID: 31669527 DOI:10.1016/j.gofs.2019.10.017

4- The EMA organizations recommend the use of Oxitocyn after almost 6 hours after the Propess removal. Why the authors decided the Ox administration only 30 minutes after the removal. 

According to manufacturer’s instructions and clinical guidelines for the IOL, the use of oxytocin is authorized 30min after device removal.

5- Why authors used theese criteria? The Triple I criteria were the most used now and worldwide.

  • At the time the protocol was written, Newton's criteria were still used. In accordance with the Triple I criteria that are now used worldwide we precised criteria of clinical chorioamnionitis in the revised manuscript (141-143).

6-Please better explain "materno-fetal infection"? Please include in the placenta hystopatology analisys the presence of"funisitis" that is more sensitive for intramniotic infection than chorioamnionitis alone because of its aspecificity.

  • In total, we reported 1 case of confirmed and 2 cases of probable maternal-fetal infections. Probable maternal-fetal infection was defined by : positive bacterial culture for gastric liquid, external ear, umbilicus, anus and/or clinical signs of infection, abnormal CRP, abnormal blood cell counts. Confirmed maternal-fetal infection was defined by bacteremia or meningitis.
  • We added the presence of funisitis in the Table 5 (3 funisitis were reported in the experimental group).
  •  
  • 7- Please clarify the indication of use of antibiotics in patients screened and negative for GBS? Is the prophylactic use of antibiotics performed in order to minimize the risk of infection due to the Baloon in women with PROM? This concept must been speculated in the discussion section.
  • In our study, patients underwent spontaneous rupture of membranes at least 12 hours before randomization.

According to french national guidelines « In the absence of spontaneous labor within 12hours of rupture, antibiotic prophylaxis could reduce the risk of maternal intrauterine infection but not of neonatal infection (LE3). Its use after 12hours of rupture in term prelabor rupture of the membranes is therefore recommended (Grade C). When antibiotic prophylaxis is indicated, intravenous beta-lactams are recommended (Grade C). »

The reference was added line 96.

[1]Term Prelabor Rupture of Membranes: CNGOF Guidelines for Clinical Practice. M-V Senat, T Schmitz, H Bouchghoul, C Diguisto, A Girault, S Paysant, J Sibiude, L Lassel, L Sentilhes PMID: 31669527 DOI:10.1016/j.gofs.2019.10.017

8- About this concept please cite the consequences of the systematic use of antibiotics in women without indication. (Microbioma, Multi resistance...)

  • In our study, the use of antibiotics was recommended for all participant because women underwent spontaneous rupture of membranes for at least 12 hours [1].

[1]Term Prelabor Rupture of Membranes: CNGOF Guidelines for Clinical Practice. M-V Senat, T Schmitz, H Bouchghoul, C Diguisto, A Girault, S Paysant, J Sibiude, L Lassel, L Sentilhes PMID: 31669527 DOI:10.1016/j.gofs.2019.10.017

Reviewer 2 Report

This is an interesting paper with high impact of its clinically relevant results. However, some questions have to be answered:

Introduction, line 42: ...when the cervix is unvaforable for labour induction.

Material and Methods, line 64: How was the diagnosis of PROM verified?

line 67: How do the authors interpret the recommendation of many international guidelines not to perform vaginal examinations in cases of PROM in connection to their study?

line 73: Why were patients excluded if they had GBS detected in "any previous pregnancy"?

line 91: What is the background for starting oxytocin 6 hours after insertion?

line 94: What do the authors mean by "expulled"?, was the insert lost by itself and another was inserted or was it removed?

line 121: "ablation" here propably means "removal"?

Results, line 163 typing error

line 163, concerning patient selection: The absolute number of PROM patients at term within this time interval has to be provided, followed by the numbers and reasons of exclusion to avoid selection bias.

Table 3 Postpartum haemorrhage: The observed postpartum hemorrhage rate with 12,5% and 15% respectively is very high in this study. PPH is defined in this study with blood loss >500ml, not differentiating between vaginal and cesarean births and not looking for cardiovascular parameters, could this explain the high numbers?

Discussion, line 240: ...however were increased...

Author Response

Response to Reviewer 2 Comments

1- when the cervix is unvaforable for labour induction.

  • Cervix was considered unfavorable for labour induction when Bishop score was < 6. This point was specified in the methods part of the revised manuscript (line 70).

2-How was the diagnosis of PROM verified?

  • Diagnosis of PROM was based on amniotic fluid leakage through the cervix observed during speculum examination. For equivocal cases, a positive detection of IGFBP-1 protein in vaginal secretions was made. This point was specified in the revised manuscript (line 70).

3- How do the authors interpret the recommendation of many international guidelines not to perform vaginal examinations in cases of PROM in connection to their study?

  • According to many recommendations this context of PROM justifies to avoid repetitive vaginal examination. Our study did not influence this precaution.

4-Why were patients excluded if they had GBS detected in "any previous pregnancy"?

  • We chose to exclude patients if they had GBS detected during the current or any previous pregnancy for this pilot study because of the recurrence of GBS colonization in successive pregnancies : recurrent GBS colonization between pregnancies is usually reported at rates of 38% to 53%.

L C Colicchia, D S Lauderdale, H Du, M Adams & E Hirsch Recurrence of group B streptococcus colonization in successive pregnancies. Journal of Perinatology 2015 ;volume 35, pages173–176.

Cheng PJ, Chueh HY, Liu CM, Hsu JJ, Hsieh TT, Soong YK . Risk factors for recurrence of group B streptococcus colonization in a subsequent pregnancy. Obstet Gynecol 2008; 111: 704–709.

Tam T, Bilinski E, Lombard E . Recolonization of group B Streptococcus (GBS) in women with prior GBS genital colonization in pregnancy. J Matern Fetal Neonatal Med 2012; 25: 1987–1989.

Turrentine MA, Ramirez MM . Recurrence of group B streptococci colonization in subsequent pregnancy. Obstet Gynecol 2008; 112: 259–264.

Page-Ramsey SM, Johnstone SK, Kim D, Ramsey PS . Prevalence of group B Streptococcus colonization in subsequent pregnancies of group B Streptococcus-colonized versus noncolonized women. Am J Perinatol 2013; 30: 383–388.

5-What is the background for starting oxytocin 6 hours after insertion?

The combined approach (Foley+ oxytocine) is frequently used in the litterature but we had no previous personal experience as this combined approach is currently not promoted in France. Therefore we opted to start oxytocin  after a delay (6 hours) to avoid excessive use of oxytocine for patients with entry into labour in the first 6 hours after balloon insertion.

Mackeen AD, Durie DE, Lin M, Huls CK, Qureshey E, Paglia MJ, et al. Foley plus oxytocin compared with oxytocin for induction after membrane rupture: A randomized controlled trial. Obstetrics and Gynecology. 1 janv 2018;131(1):4‑11.

Orr L, Reisinger-Kindle K, Roy A, Levine L, Connolly K, Visintainer P, Schoen

  1. Combination of Foley and prostaglandins versus Foley and oxytocin for

cervical ripening: a network meta-analysis. Am J Obstet Gynecol. 2020

Nov;223(5):743.e1-743.e17. doi: 10.1016/j.ajog.2020.05.007. Epub 2020 May 7.

PMID: 32387325.

Gagnon J, Corlin T, Berghella V, Hoffman MK, Sciscione A, Marie PS, Schoen

  1. Intracervical Foley catheter with and without oxytocin for labor induction

with Bishop score ≤3: a secondary analysis. Am J Obstet Gynecol MFM. 2021

Jul;3(4):100350. doi: 10.1016/j.ajogmf.2021.100350. Epub 2021 Mar 20. PMID:

33757937.

6-What do the authors mean by "expulled"?, was the insert lost by itself and another was inserted or was it removed?

  • Expulsed was replaced by « lost by itself » in the revised manuscript (line107)

In cases where the dinoprostone insert was lost by itself in the first 12 hours and the patient had no contractions or continued to present an unfavorable cervix, another vaginal system was placed for a maximum further 24 hours.

7-"ablation" here propably means "removal"?

  • « ablation » was replaced by « removal » (Lines 138 and 141)

8-line 190 : typing error was corrected

9-Concerning patient selection: The absolute number of PROM patients at term within this time interval has to be provided, followed by the numbers and reasons of exclusion to avoid selection bias.

lines 191-192: the following  sentence was added : « From February 2018 to March 2019, … Over this period the refusal rate was 27%, reason for refusal was related to randomization. »

10-Table 3 Postpartum haemorrhage: The observed postpartum hemorrhage rate with 12,5% and 15% respectively is very high in this study. PPH is defined in this study with blood loss >500ml, not differentiating between vaginal and cesarean births and not looking for cardiovascular parameters, could this explain the high numbers?

  • We chose to present PPH rates according to french guidelines, indicating that regardless of the route of delivery, postpartum haemorrhage (PPH) is defined as blood loss ≥ 500 mL after delivery (without differentiating vaginal and cesarean birth). The definition was added in the revised manuscript to clarify this point (line 250). In population-based studies, the incidence of PPH is around 10% when blood loss is quantified. Our rate of PPH is slightly higher but we are not able to explain this difference with our data. Labour induction is usually accepted as a risk factor for PPH.
